# Bounded-Loss Private Prediction Markets

**Rafael Frongillo**
Colorado Boulder
raf@colorado.edu

**Bo Waggoner**
Microsoft Research
bwag@colorado.edu

## Abstract

Prior work has investigated variations of prediction markets that preserve participants' (differential) privacy, which formed the basis of useful mechanisms for purchasing data for machine learning objectives. Such markets required potentially unlimited financial subsidy, however, making them impractical. In this work, we design an adaptively-growing prediction market with a bounded financial subsidy, while achieving privacy, incentives to produce accurate predictions, and precision in the sense that market prices are not heavily impacted by the added privacy-preserving noise. We briefly discuss how our mechanism can extend to the data-purchasing setting, and its relationship to traditional learning algorithms.

## 1 Introduction

In a prediction market, a platform maintains a prediction (usually a probability distribution or an expectation) of a future random variable such as an election outcome. Participants' trades of financial securities tied to this event are translated into updates to the prediction. Prediction markets, designed to aggregate information from participants, have gained a substantial following in the machine learning literature. One reason is the overlap in goals (predicting future outcomes) as well as techniques (convex analysis, Bregman divergences), even at a deep level: the form of market updates in standard automated market makers have been shown to mimic standard online learning or optimization algorithms in many settings [2, 3, 8, 9]. Beyond this research-level bridge, recent papers have suggested prediction market mechanisms as a way of *crowdsourcing* data or algorithms for machine learning, usually by providing incentives for participants to repeatedly update a centralized hypothesis or prediction [4, 12].

One recently-proposed mechanism to purchase data or hypotheses from participants is that of Waggoner, et al. [12], in which participants submit updates to a centralized market maker, either by directly altering the hypothesis, or in the form of submitted data; both are interpreted as buying or selling shares in a market, paying off according to a set of holdout data that is revealed after the close of the market. The authors then show how to preserve *differential privacy* for participants, meaning that the content of any individual update is protected, as well as natural accuracy and incentive guarantees.

One important drawback of Waggoner, et al. [12], however, is the lack of a *bounded worst-case loss* guarantee: as the number of participants grows, the possible financial liability of the mechanism grows without bound. In fact, their mechanism cannot achieve a bounded worst-case loss without giving up privacy guarantees. Subsequently, Cummings, et al. [7] show that *all* differentially-private prediction markets of the form proposed in [12] must suffer from unbounded financial loss in the worst case. Intuitively, one could interpret this negative result as saying that the randomness of the mechanism, which must be introduced to preserve privacy, also creates *arbitrage* opportunities for participants: by betting against the noise, they collectively expect to make an unbounded profit from the market maker. Nevertheless, Cummings, et al. leave open the possibility that mechanisms outside the mold of Waggoner, et al. could achieve both privacy and a bounded worst-case loss.

In this paper, we give such a mechanism: the first private prediction market framework with a bounded worst-case loss. When applied to the crowdsourcing problems stated above, this now allows the mechanism designer to maintain a fixed budget. Our construction and proof proceeds in two steps.

We first show that by adding a small transaction fee to the mechanism of [12], one can eliminate financial loss due to arbitrage while maintaining the other desirable properties of the market. The key idea is that a carefully-chosen transaction fee can make each trader subsidize (in expectation) any arbitrage that may result from the noise preserving her privacy. Unless prices already match her beliefs quite closely, however, she still expects to make a profit by paying the fee and participating. We view this as a positive result both conceptually—it shows that arbitrage opportunities are not an insurmountable obstacle to private markets—and technically—the designer budget grows very slowly, only $O((\log T)^2)$, with the number of participants $T$.

Nonetheless, this first mechanism is still not completely satisfactory, as the budget is superconstant in $T$, and $T$ must be known in advance. This difficulty arises not from arbitrage, but (apparently) a deeper constraint imposed by privacy that forces the market to be large relative to the participants. Our second and main result overcomes this final hurdle. We construct a sequence of adaptively-growing markets that are syntactically similar to the "doubling trick" in online learning. The key idea is that, in the market from our first result, only about $(\log T)^2$ of the $T$ participants can be *informational* traders; after this point, additional participants do not cost the designer any more budget, yet their transaction fees can raise significant funds. So if the end of a stage is reached, the market activity has actually generated a surplus which subsidizes the initial portion of the next stage of the market.

## 2  Setting

In a cost-function based prediction market, there is an observable future outcome $Z$ taking values in a set $\mathcal{Z}$. The goal is to predict the expectation of a random variable $\phi : \mathcal{Z} \to \mathbb{R}^d$. We assume $\phi$ is a bounded random variable, as otherwise prediction markets with bounded financial loss are not possible. Participants will buy from the market *contracts*, each parameterized by a vector $r \in \mathbb{R}^d$. The contract represents a promise for the market to pay the owner $r \cdot \phi(Z)$ when $Z$ is observed. Adopting standard financial terminology, in our model there are $d$ *securities* $j = 1, \ldots, d$, and the owner of a *share* in security $j$ will receive a payoff of $\phi(Z)_j$, that is, the $j$th component of the random variable. Thus a contract $r \in \mathbb{R}^d$ contains $r_j$ shares of security $j$ and pays off $\sum_{j=1}^{d} r_j \phi(Z)_j = r \cdot \phi(Z)$. Note that $r_j < 0$, or "short selling" security $j$, is allowed.

The market maintains a *market state* $q^t \in \mathbb{R}^d$ at time $t = 0, \ldots, T$, with $q^0 = 0$. Each trader $t = 1, \ldots, T$ arrives sequentially and purchases a contract $dq^t \in \mathbb{R}^d$, and the market state is updated to $q^t = q^{t-1} + dq^t$. In other words, $q^t = \sum_{s=1}^{t} dq^s$, the sum of all contracts purchased up to time $t$. The price paid by each participant is determined by a convex *cost function* $C : \mathbb{R}^d \to \mathbb{R}$. Intuitively, $C$ maps $q^t$ to the total price paid by all agents so far, $C(q^t)$. Thus, participant $t$ making trade $dq^t$ when the current state is $q^{t-1}$ pays $C(q^{t-1} + dq^t) - C(q^{t-1})$. Notice that the *instantaneous prices* $p^t = \nabla C(q^t)$ represent the current price per unit of infinitesimal purchases, with the $j$th coordinate representing the current price per share of the $j$th security.

The prices $\nabla C(q)$ are interpreted as predictions of $\mathbb{E}\,\phi(Z)$, as an agent who believes the $j$th coordinate is too low will purchase shares in it, raising its price, and so on. This can be formalized through a learning lens: It is known [2] that agents in such a market maximize expected profit by minimizing an expected Bregman divergence between $\phi(Z)$ and $\nabla C(q)$; of course, it is known that $\nabla C(q) = \mathbb{E}\,\phi(Z)$ minimizes risk for any divergence-based loss [1, 6, 10]. (The Bregman divergence is that corresponding to $C^*$, the convex conjugate of $C$.)

**Price Sensitivity.**  The price sensitivity of a cost function $C$ is a measure of how quickly prices respond to trades, similar to "liquidity" discussed in Abernethy et al. [2, 5] and earlier works. Formally, the *price sensitivity* $\lambda$ of $C$ is the supremum of the operator norm of the Hessian of $C$, with respect to the $\ell_1$ norm.[1] In other words, if $c = \|q - q'\|_1$ shares are purchased, then the change in prices $\|\nabla C(q) - \nabla C(q')\|_1$ is at most $\lambda c$.

Price sensitivity is directly related to the worst-case loss guarantee of the market, as follows. Those familiar with market scoring rules may recall that with scoring rule $S$, the loss can be bounded by (a constant times) the largest possible score. Hence, scaling $S$ by a factor $\frac{1}{\lambda}$ immediately scales the loss bound by $\frac{1}{\lambda}$ as well. Recall that $S$ is defined by a convex function $G$, the convex conjugate of $C$. Scaling $S$ by $\frac{1}{\lambda}$ is equivalent to scaling $G$ by $\frac{1}{\lambda}$. By standard results in convex analysis, this is equivalent to transforming $C$ into $C_\lambda(q) = \frac{1}{\lambda} C(\lambda q)$, an operation known as the perspective transform. This in turn scales the price sensitivity by $\lambda$ by the properties of the Hessian.

Price sensitivity is also related to the total number of trades required to change the prices in a market. If we assume each trade consists of at most one share in each security, then $\frac{1}{\lambda}$ trades are necessary to shift the predictions to an arbitrary point from an arbitrary point.

**Convention: normalized, scaled $C$.** In the remainder of the paper, we will suppose that we start with some convex cost function $C_1$ whose price sensitivity equals 1 and worst-case loss bounded by some constant $B_1$. Then, to obtain price sensitivity $\lambda$, we use the cost function $C(\cdot) = \frac{1}{\lambda} C_1(\lambda \cdot)$. As discussed above, $C$ has price sensitivity at most $\lambda$ and a worst-case loss bound of $B = B_1/\lambda$. (This assumption is without loss of generality, as any cost function that guarantees a bounded worst-case loss can be scaled such that its price sensitivity is 1.)

## 2.1 Prior work

To achieve differential privacy for trades of a bounded size (which will be assumed), the general approach is to add random noise to the "true" market state $q$ and publish this noisy state $\hat{q}$. The privacy level thus determines how close $\hat{q}$ is to $q$. The distance from $\nabla C(\hat{q})$ to $\nabla C(q)$ is then controlled by the price sensitivity $\lambda$. For a fixed noise and privacy level, a smaller $\lambda$ leads to small impact of noise on prices, meaning very good accuracy. However, decreasing $\lambda$ does not come for free: the worst-case financial loss of to the market designer scales as $1/\lambda$.

The market of [12] adds controlled and correlated noise over time, in a manner similar to the "continual observation" technique of differential privacy. This reduces the influence of noise on accuracy to polylogarithmic in $T$, the number of participants. Their main result for the prediction market setting studied here is as follows.

**Theorem 1** ([12]). *Assuming that all trades satisfy $\|dq^t\|_1 \leq 1$, the private mechanism is $\epsilon$-differentially private in the trades $dq^1, \ldots, dq^T$ with respect to the output $\hat{q}^1, \ldots, \hat{q}^T$. Further, to satisfy $\|p^t - \hat{p}^t\|_1 \leq \alpha$ for all $t$, except with probability $\gamma$, it suffices for the price sensitivity to be*

$$\lambda^* = \frac{\alpha\,\epsilon}{4\sqrt{2}d\lceil \log T \rceil \ln(2Td/\gamma)}\;. \tag{1}$$

## 2.2 Our setting and desiderata

This paper builds on the work of Waggoner et al. [12] to overcome the negative results of Cummings et al. [7]. Here, we formalize our setting and four desirable properties we hope to achieve.

Write a prediction market mechanism as a function $M$ taking inputs $\vec{dq} = dq^1, \ldots, dq^T$ and outputting a sequence of market states $\hat{q}^1, \ldots, \hat{q}^T$. Here $\hat{q}^t$ is thought of as a noisy version of $q^t = \sum_{s \leq t} dq^s$. Each of these states is associated with a prediction $\hat{p}^t$ in the set of possible prices (expectations of $\phi$), while the state $q^t$ is associated with the "true" underlying prediction $p^t$.

**Definition 1** (Privacy). *$M$ satisfies $(\epsilon, \delta)$-differential privacy if for all pairs of inputs $\vec{dq}, \vec{dq}'$ differing by only a single participants' entry, and for all sets $S$ of possible outputs, $\Pr[M(\vec{dq}) \in S] \leq e^\epsilon \Pr[M(\vec{dq}') \in S] + \delta$. If furthermore $\delta = 0$, we say $M$ is $\epsilon$-differentially private.*

**Definition 2** (Precision). *$M$ has $(\alpha, \gamma)$ precision if for all $\vec{dq}$, with probability $1 - \gamma$, we have $\|\hat{p}^t - p^t\|_1 \leq \alpha$ for all $t$.*

**Definition 3** (Incentives). *$M$ has $\beta$-incentive to participate if, for all beliefs $p = \mathbb{E}\,\phi(Z)$, if at any point $\|\hat{p}^t - p\|_\infty > \beta$, then there exists a participation opportunity that makes a strictly positive profit in expectation with respect to $p$.*

For the budget guarantee, we must formalize the notion that participants may respond to the noise introduced by the mechanism. Following Cummings et al. [7], let a *trader*

*strategy* $\vec{s} = (s^1, \ldots, s^T)$ where each $s^t$ is a possibly-randomized function of the form $s^t(dq^1, \ldots, dq^{t-1}; \hat{q}^1, \ldots, \hat{q}^{t-1}) = dq^t$, i.e. a strategy taking the entire history prior to $t$ and outputting a trade $dq^t$. Let $L(M, \vec{s}, z)$ be a random variable denoting the financial loss of the market $M$ against trader strategy $\vec{s}$ when $Z = z$, which for the mechanism described above is simply

$$L(M, \vec{s}, z) = \sum_{t=1}^{T} \left[ C(\hat{q}^t) - C(\hat{q}^t + dq^t) - dq^t \cdot \phi(z) \right].$$

**Definition 4.** $M$ guarantees *designer budget* $B$ if, for any trader strategy $\vec{s}$ and all $z$, $\mathbb{E}\, L(M, \vec{s}, z) \leq B$, where the expectation is over the randomness in $M$ and each $s^t$.

## 3  Slowly-Growing Budget

The private market of Waggoner et al. [12] causes unbounded loss for the market maker in two ways. The first is from traders betting against the random noise introduced to protect privacy. This is a key idea leveraged by Cummings et al. [7] to show negative results for private markets. In this section, we show that a transaction fee can be chosen to exactly balance the expected profit from this type of arbitrage.[2] We will show that this fee is still small enough to allow for very accurate prices.[3] This transaction fee restores the worst-case loss guarantee to the inverse of the price sensitivity, just as in a non-private market. The second way the market causes unbounded loss is to require price sensitivity to shrink as a function of $T$; this is addressed in the next section.

We show that with this carefully-chosen fee, the market still achieves precision, incentive, and privacy guarantees, but now with a worst-case market maker loss of $O((\log T)^2)$, much improved over the naïve $O(T)$ bound. This is viewed as a positive result because the worst-case loss is growing quite slowly in the total number of participants, and moreover matches the fundamental "informational" worst-case loss one expects with price sensitivity $\lambda^*$.

### 3.1  Mechanism and result

Here we recall the private market mechanism of [12], adapted to the prediction market setting following [7]. We will express the randomness of the mechanism in terms of a "noise trader" for both intuition and technical convenience. The market is defined by a cost function $C$ with price sensitivity $\lambda$, and parameters $c$ (transaction fee), $\epsilon$ (privacy), $\alpha, \gamma$ (precision), and $T$ (maximum number of participants). There is a special trader we call the *noise trader* who is controlled by the designer. All actions of the noise trader are hidden and known only by the designer. The designer publishes an initial market state $q^0 = \hat{q}^0 = 0$. Let $T'$ denote the actual number of arrivals, with $T' \leq T$ by assumption. Then, for $t = 1, \ldots, T'$:

1. Participant $t$ arrives, pays a fee of $c$, and purchases bundle $dq^t$ with $\|dq^t\|_1 \leq 1$. The payment is $C(\hat{q}^t + dq^t) - C(\hat{q}^t)$.
2. The noise trader purchases a randomly-chosen bundle $z^t$, called a noise trade, after selling off some subset $\{z^{t_1}, \ldots, z^{t_k}\}$ of previously purchased noise trades for $t_i < t$, according to a predetermined schedule described below. Letting $w^t = z^t - \sum_{i=1}^{k} z^{t_i}$ denote this net noise bundle, the noise trader is thus charged $C(\hat{q}^t + dq^t + w^t) - C(\hat{q}^t + dq^t)$.
3. The "true" market state is updated to $q^t = q^{t-1} + dq^t$, but is not revealed.
4. The noisy market state is updated to $\hat{q}^t = \hat{q}^{t-1} + dq^t + w^t$ and is published.

Finally, $z \in \mathcal{Z}$ is observed and each participant $t$ receives a payment $dq^t \cdot \phi(z)$. For the sake of budget analysis, we suppose that at the close of the market, the noise trader sells back all of her remaining bundles; letting $w^{T'}$ be the sum of these bundles, she is charged $C(\hat{q}^{T'} - w^{T'}) - C(\hat{q}^{T'})$.

**Noise trades.** Each $z^t$ is a $d$-dimensional vector with each coordinate drawn from an independent Laplace distribution with parameter $b = 2\lceil \log T \rceil / \epsilon$. To determine which bundles $z^s$ are sold at time $t$, write $t = 2^j m$ where $m$ is odd, and sell all bundles $z^s$ purchased during the previous

$2^{j-1}$ time steps which are not yet sold. Thus, the noise trader will sell bundles purchased at times $s = t - 1, t - 2, t - 4, t - 8, \ldots, t - 2^{j-1}$; in particular, when $t$ is odd we have $j = 0$, so no previous bundles will be sold.

**Budget.** The total loss of the market designer can now be written as the sum of three terms: the loss of the market maker, the loss of the noise trader, and the gain from transaction fees. By convention, the noise trader eventually sells back all bundles it purchases and is left with no shares remaining.

$$L(M, \vec{s}, z) = \overbrace{\sum_{t=1}^{T'} C(\hat{q}^{t-1}) - C(\hat{q}^{t-1} + dq^t) + dq^t \cdot \phi(z)}^{\text{net loss of market maker}} + \overbrace{\sum_{t=1}^{T'} C(\hat{q}^{t-1} + dq^t) - C(\hat{q}^t)}^{\text{net loss of noise trader}} + \overbrace{cT'}^{\text{fees}}. \quad (2)$$

The main result of this section is as follows.

**Theorem 2.** *When each arriving participant pays a transaction fee $c = \alpha$, the private market with any $\lambda \leq \lambda^*$ from eq. (1) satisfies $\epsilon$-differential privacy, $(\alpha, \gamma)$-precision, $2\alpha$-incentive to trade, and budget bound $\frac{B_1}{\lambda}$, where $B_1$ is the budget bound of the underlying cost function $C_1$.*

### 3.2 Proof ideas: privacy, precision, incentives

The differential privacy and precision claims follow directly from the prior results, as nothing has changed to impact them. The incentive claim is not technically involved, but perhaps subtle: the transaction fee should be high enough to eliminate expected profit from arbitrage, yet low enough to allow for profit from information. The key point is that the transaction fee is a constant, but the farther the prices are from the trader's belief, the more money she expects to make from a constant-sized trade. The transaction fee creates a ball of size $2\alpha$ around the current prices where, if one's belief lies in that ball, then participation is not profitable.

We give most of the proof of the designer budget bound, with some claims deferred to the full version.

**Lemma 1** (Budget bound). *The transaction-fee private market with any price sensitivity $\lambda \leq \lambda^*$ guarantees a designer budget bound of $\frac{B_1}{\lambda}$.*

*Proof.* Let $c$ be the transaction fee; we will later take $c = \alpha$. Then the worst-case loss from eq. (2) is

$$WC(\lambda, T') := WC_0(\lambda, T') + NTL(\lambda, T') - T'c,$$

where $WC_0(\lambda, T')$ is the worst-case loss of a standard prediction market maker with parameter $\lambda$ and $T'$ participants, $NTL(\lambda, T')$ is the worst-case noise trader loss, and $T'c$ is the revenue from $T'$ transaction fees of size $c$ each.

The worst-case loss of a standard prediction market maker is well-known; see e.g. [2]. By our normalization and definition of price sensitivity, we thus have $WC_0(\lambda, T') \leq \frac{B_1}{\lambda}$.

To bound the noise trader loss $NTL(\lambda, T')$, we will consider each bundle $z^t$ purchased by the noise trader. The idea is to bound the difference in price between the purchase and sale of $z^t$. For analysis, we suppose that at each $t$, the noise trader first sells any previous bundles (e.g. at $t = 4$, first selling $z^3$ and then selling $z^2$), and then purchases $z^t$.

Now let $b(t)$ be the largest power of 2 that divides $t$. Let $q_{\text{buy}}^t$ and $q_{\text{sell}}^t$ be the market state just before the noise trader purchases $z^t$ and just after she sells $z^t$, respectively.

**Claim 1.** *For each $t$, exactly $b(t)$ traders arrive between the purchase and the sale of bundle $z^t$; furthermore, $q_{sell}^t - q_{buy}^t$ is exactly equal to the sum of these participants' trades.*

For example, suppose $t$ is odd. Then only one participant arrives between the purchase and sale of $z^t$. Furthermore, $z^t$ is the last bundle purchased by the noise trader at time $t$ and is the first sold at time $t + 1$, so the difference in market state is exactly $z^t$ plus that participant's trade.

**Claim 2.** *If the noise trader purchases and later sells $z^t$, then her net loss in expectation over $z^t$ (but for any trader behavior in response to $z^t$), is at most $\lambda b(t) K$ where $K = \mathbb{E}\|z^t\|_2$.*

We now sum over all bundles $z^t$ purchased by the noise trader, i.e. at time steps $1, \ldots, T'$. Recall that the noise trader sells back every bundle $z^t$ she purchases. Thus, her total payoff is the sum over $t$ of the difference in price at which she buys $z^t$ and price at which she sells it. For each $j = 0, \ldots, \log T' - 1$, there are $2^j$ different steps $t$ with $b(t) = T'/2^{j+1}$. The total loss is thus,

$$NTL(\lambda, T') \leq \sum_{j=0}^{\log T'-1} 2^j \frac{T'}{2^{j+1}} \lambda K = \frac{T' \log T'}{2} \lambda K \ . \tag{3}$$

Note that if the noise trader has some noise bundles left over after the final participant, we suppose she immediately sells all remaining bundles back to the market maker in reverse order of purchase.

Putting eq. (3) together with the above bound on $WC_0$ gives

$$WC(\lambda, T') \leq WC_0(\lambda, T') + T' \log T' \lambda K - T'c \leq \frac{B_1}{\lambda} + T' \left( K \log T' \lambda - c \right) \ , \tag{4}$$

which is in turn at most $B_1/\lambda$ if we choose $\lambda$ and the transaction fee $c$ such that $c \geq K \log T \lambda$. In other words, we take $\lambda \leq c/K \log T$.

Finally, we can bound $K = \mathbb{E} \|z^t\|_2$ from Claim 2 as follows: for each $t$, the components of the $d$-dimensional vector $z^t$ are each independent $\mathrm{Lap}(b)$ variables with $b = 2\lceil \log T \rceil/\epsilon$. By concavity of $\sqrt{\cdot}$, we have

$$K = \mathbb{E} \sqrt{\sum_{i=1}^{d} z^t(i)^2} \leq \sqrt{\sum_i \mathbb{E} \, z^t(i)^2} = \sqrt{d \mathrm{Var}(\mathrm{Lap}(b))} = \sqrt{2db^2} = 2\sqrt{2d} \frac{\lceil \log T \rceil}{\epsilon} \ .$$

Therefore, it suffices to pick

$$\lambda \leq \frac{c \, \epsilon}{2\sqrt{2d} \lceil \log T \rceil \log T} \ .$$

For $c = \alpha$, this is in fact accomplished by the private, accurate market choosing $\lambda \leq \lambda^*$ (Equation 1). $\qquad\square$

**Limitations of this result.** Unfortunately, Theorem 2 does not completely solve our problem: the other way that privacy impacts the market's loss is by lowering the necessary price sensitivity to $\lambda^* \approx \frac{1}{(\log T)^2}$ as mentioned above, leading to a worst-case loss growing with $T$. It does not seem possible to address this via a larger transaction fee without giving up incentive to participate: traders participate as long as their expected profit exceeds the fee, and collectively $\Omega(1/\lambda)$ of them can arrive making consistent trades all moving the prices in the same (correct) direction, so the total payout will still be $\Omega(1/\lambda)$.

## 4 Constant Budget via Adaptive Market Size

In this section, we achieve our original goal by constructing an adaptively-growing prediction market in which each stage, if completed, subsidizes the initial portion of the next.

The market design is the following, with each $T^{(k)}$ to be chosen later. We run the transaction-fee private market above with $T = T^{(1)}$, transaction fee $\alpha$, and price sensitivity $\lambda^{(1)} = \lambda^*(T^{(1)}, \alpha/2, \gamma/2)$ from eq. (1). When (and if) $T^{(1)}$ participants have arrived, we create a new market whose initial state is such that its prices match the final (noisy) prices of the previous one. We set $T^{(2)}$ and price sensitivity $\lambda^{(2)} = \lambda^*(T^{(2)}, \alpha/4, \gamma/4)$ for the new market. We repeat, halving $\alpha$ and $\gamma$ at each stage and increasing $T$ in a manner to be specified shortly, until no more participants arrive.

**Theorem 3.** *For any $\alpha, \gamma, \epsilon$, the adaptive market satisfies $\epsilon$-differential privacy, $2\alpha$-incentive to trade, $(\alpha, \gamma)$-accuracy, and a designer budget bound of*

$$B \leq B_1 \frac{72\sqrt{2}d}{\alpha \, \epsilon} \left( \ln \frac{4608 B_1 \sqrt{2} d^2}{\gamma \alpha^2 \epsilon} \right)^2 ,$$

*where $B_1$ is the budget bound of the underlying unscaled cost function $C_1$.*

**Proof idea.** We set $T^{(1)} = \Theta\left(\frac{B_1 d \ln(B_1 d/\gamma\alpha\epsilon)^2}{\alpha^2} \frac{1}{\epsilon}\right)$, and $T^{(k)} = 4T^{(k-1)}$ thereafter. The key will be the following observation. The total "informational" profit available to the traders (by correcting the initial market prices) is bounded by $O(1/\lambda)$, so if each trader expects to profit more than the transaction fee $c$, then only $O(1/\lambda c)$ traders can all arrive and simultaneously profit. Indeed, if all $T$ participants arrive, then the total profit from transaction fees is $\Theta(T)$ while the worst-case loss from the market is $O\left((\log T)^2\right)$.

We can leverage this observation to achieve a bounded worst-case loss with an "adaptive-liquidity" approach, similar in spirit to Abernethy et al. [5] but more technically similar to the doubling trick in online learning. Begin by setting $\lambda^{(1)}$ on the order of $1/(\log T^{(1)})^2 = \Theta(1)$, and run a private market for $T^{(1)}$ participants. If fewer than $T^{(1)}$ participants show up, the worst-case loss is order $1/\lambda^{(1)}$, a constant. If all $T^{(1)}$ participants arrive, then (for the right choice of constants) the market has actually turned a profit $\Omega(T^{(1)})$ from the transaction fees. Now set up a private market for $T^{(2)} = 4T^{(1)}$ traders with $\lambda^{(2)}$ on the order of $1/(\log T^{(2)})^2$. If fewer than $T^{(2)}$ participants arrive, the worst-case loss is order $1/\lambda^{(2)}$. However, we will have chosen $T^{(2)}$ such that this loss is smaller than the $\Omega(T^{(1)})$ profit from the previous market. Hence, the total worst-case loss remains bounded by a constant.

If all $T^{(2)}$ participants arrive, then again this market has turned a profit, which can be used to completely offset the worst-case loss of the next market, and so on. Some complications arise, as to achieve $(\alpha, \gamma)$-precision, we must set $\alpha^{(1)}, \gamma^{(1)}, \alpha^{(2)}, \gamma^{(2)}, \ldots$ as a convergent series summing to $\alpha$ and $\gamma$; and we must show that all of these scalings are possible in such a way that the transaction fees cover the cost of the next iteration. (An interesting direction for future work would be to replace the iterative approach here with the continuous liquidity adaptation of [5].)

More specifically, we prove that the loss in any round $k$ that is not completed (not all participants arrive) is at most $\frac{\alpha}{16}T^{(k)}$; moreover, the profit in any round $k$ that is completed is at least $\frac{\alpha}{2}T^{(k)}$. Of course, only one round is not completed: the final round $k$. If $k = 1$, then the financial loss is bounded by $\frac{1}{\lambda^{(1)}}$, a constant depending only on $\alpha, \gamma, \epsilon$. Otherwise, the total loss is the sum of the losses across rounds, but the mechanism makes a profit in every round but $k$. Moreover, the loss in round $k$ is at most $\frac{\alpha}{2}T^{(k)} = \frac{\alpha}{8}T^{(k-1)}$, which is at most half of the profit in round $k-1$. So if $k \geq 2$, the mechanism actually turns a net profit.

While this result may seem paradoxical, note that the basic phenomenon appears in a classical (non-private) prediction market with a transaction fee, although to our knowledge this observation has not yet appeared in the literature. Specifically, a classical prediction market with budget bound $B_1$, trades of size 1, and a small transaction fee $\alpha$, will still have an $\alpha$-incentive to participate, and the worst case loss will still be $\Theta(B_1)$; this loss, however, can be extracted by as few as $\Theta(1)$ participants. Any additional participants must be in a sense disagreeing about the correct prices; their transaction fees go toward market maker profit, but they do not contribute further to worst-case loss.

## 5   Kernels, Buying Data, Online Learning

While preserving privacy in prediction markets is well-motivated in the classical prediction market setting, it is arguably even more important in a setting where machine-learning hypotheses are learned from private personal data. Waggoner et al. [12] develop mechanisms for such a setting based on prediction markets, and further show how to preserve differential privacy of the participants. Yet their mechanisms are not practical in the sense that the financial loss of the mechanism could grow without bound. In this section, we sketch how our bounded-financial-loss market can also be extended to this setting. This yields a mechanism for purchasing data for machine learning that satisfies $\epsilon$-differential privacy, $\alpha$-precision and incentive to participate, and bounded designer budget.

To develop a mechanism which could be said to "purchase data" from participants, Waggoner et al. [12] extend the classical setting in two ways. The first is to make the market *conditional*, where we let $\mathcal{Z} = \mathcal{X} \times \mathcal{Y}$, and have independent markets $C_x : \mathbb{R}^d \to \mathbb{R}$ for each $x$. Trades in each market take the form $q_x \in \mathbb{R}^d$, which pay out $q_x \cdot \phi(y)$ upon outcome $(x', y)$ if $x = x'$, and zero if $x \neq x'$. Importantly, upon outcome $(x, y)$, only the costs associated to trades in the $C_x$ market are tallied.

The second is to change the bidding language using a *kernel*, a positive semidefinite function $k : \mathcal{Z} \times \mathcal{Z} \to \mathbb{R}$. Here we think of contracts as functions $f : \mathcal{Z} \to \mathbb{R}$ in the reproducing kernel Hilbert space (RKHS) $\mathcal{F}$ given by $k$, with basis $\{f_z(\cdot) = k(z, \cdot) \; : \; z \in \mathcal{Z}\}$. For example, we recover

the conditional market setting with independent markets with the kernel $k((x,y),(x',y')) = \mathbf{1}\{x = x'\}\phi(y) \cdot \phi(y')$. The RKHS structure is natural here because a basis contract $f_z$ pays off at each $z'$ according to the "covariance" structure of the kernel, i.e. the payoff of contract $f_z$ when $z'$ occurs equals $f_z(z') = k(z, z')$. For example, when $\mathcal{Y} = \{-1, 1\}$ one recovers radial basis classification using $k((x,y),(x',y')) = yy'e^{-(x-x')^2}$.

These two modifications to classical prediction markets, given as Mechanism 2 in [12], have clear advantages as a mechanism to "buy data". One may imagine that each agent, arriving at time $t \in \{1, \ldots, T\}$, holds a data point $(x^t, y^t) \in \mathcal{Z} = \mathcal{X} \times \mathcal{Y}$. A natural purchase for this agent would be a basis contract $f_{(x^t, y^t)}$, as this corresponds to a payoff that is highest when the test data point actually equals $(x^t, y^t)$ and decreases with distance as measured by the kernel structure.

The importance of privacy now becomes even more apparent, as the data point $(x^t, y^t)$ could be information sensitive to trader $t$. Fortunately, we can extend our main results to this setting. To demonstrate the idea, we give a sketch of the result and proof below.

**Theorem 4** (Informal). *Let $\mathcal{Z} = \mathcal{X} \times \mathcal{Y}$ where $\mathcal{X}$ is a compact subset of a finite-dimensional real vector space and $\mathcal{Y}$ is finite, and let positive semidefinite kernel $k : \mathcal{Z} \times \mathcal{Z} \to \mathbb{R}$ be given. For any choices of accuracy parameters $\alpha, \gamma$, privacy parameters $\epsilon, \delta$, trade size $\Delta$, and query limit $Q$, the kernel adaptive market satisfies $(\epsilon, \delta)$-differential privacy, $(\alpha, \gamma)$-precision, $2\alpha$-incentive to participate, and a bounded designer budget.*

*Proof Sketch.* The precision property, i.e. that prices are approximately accurate despite privacy-preserving noise, follows from [12, Theorem 2], and the technique in Theorem 3 to combine the accuracy and privacy of multiple epochs. The incentive to trade property is essentially unchanged, as a participants' profit is still the improvement in expected Bregman divergence, which exceeds the transaction fee unless prices are already accurate. It thus remains only to show a bounded designer budget, which is slightly more involved. Briefly, Claim 1 goes through unchanged, and Claim 2 holds as written where now $C$ becomes $C_x$ and $z^t$ becomes $z^t(x) = f^t(x, \cdot)$, i.e., the trade at time $t$ restricted to the $C_x$ market alone.

The remainder of Lemma 1 now proceeds with one modification regarding the constant $K$. In eq. (3), the expression for the noise trader loss becomes $NTL(\lambda, T') = \mathbb{E}\big[\sup_{x \in \mathcal{X}} \sum_{t=1}^{T'} \lambda\alpha_t \|z^t(x)\|_2\big]$, where the $\alpha_t$ are simply coefficients to keep track of how many trades occurred between the buy and sell of noice trade $t$. We can proceed as follows:

$$NTL(\lambda, T') \leq \mathbb{E}\left[\sup_{x_1, \ldots, x_{T'} \in \mathcal{X}} \sum_{t=1}^{T'} \lambda\alpha_t \|z^t(x_t)\|_2\right] = \lambda \sum_{t=1}^{T'} \alpha_t \mathbb{E}\left[\sup_{x \in \mathcal{X}} \|z^t(x)\|_2\right] = \lambda \sum_{t=1}^{T'} \alpha_t K,$$

where $K$ is simply the constant $\mathbb{E}\left[\sup_{x \in \mathcal{X}} \|z^t(x)\|_2\right]$ where the expectation is taken over the Gaussian process generating the noise. It is well-known that the expected maximum of a Gaussian process is bounded [11], and thus boundedness of $K$ follows from the fact that $\mathcal{Y}$ is finite. Thus, continuing from eq. (3) we obtain $NTL(\lambda, T') \leq \frac{T' \log T'}{2} \lambda K$ as before, with this new $K$. Finally, the proof of Theorem 3 now goes through, as it only treats the mechanism from Theorem 2 as a black box. $\qquad\square$

We close by noting the similarity between the kernel adaptive market mechanism and traditional learning algorithms, as alluded to in the introduction. As observed by Abernethy, et al. [2], the market price update rule for classical prediction markets resembles Follow-the-Regularized-Leader (FTRL); specifically, the price update at time $t$ is given by $p^t = \nabla C(q^t) = \text{argmax}_{w \in \Delta(\mathcal{Y})} \langle w, \sum_{s \leq t} dq^s \rangle - R(w)$, where $dq^s$ is the trade at time $s$, and $R = C^*$ is the convex conjugate of $C$.

In our RKHS setting, we can see the same relationship. For concreteness, let $C_x(q) = \frac{1}{\lambda} C(\lambda q)$ for all $x \in \mathcal{X}$, and let $R : \Delta(\mathcal{Y}) \to \mathbb{R}$ be the conjugate of $C$. Suppose further that each agent $t$ purchases a basis contract $df^t = f_{x^t, y^t}$, where we take a classification kernel $k'((x,y),(x',y')) =$

$k(x, x')\mathbf{1}\{y = y'\}$. Letting $dq^t(x) = df^t(x, \cdot) \in \mathbb{R}^{\mathcal{Y}}$, the market price at time $t$ is given by,

$$
\begin{aligned}
p_x^t &= \operatorname*{argmax}_{w \in \Delta(\mathcal{Y})} \left\langle w, \sum_{s \leq t} dq^s(x) \right\rangle - \frac{1}{\lambda} R(w) \\
&= \operatorname*{argmax}_{w \in \Delta(\mathcal{Y})} \left\langle w, \sum_{s \leq t} k((x^s, y^s), (x, \cdot)) \right\rangle - \frac{1}{\lambda} R(w) \\
&= \operatorname*{argmax}_{w \in \Delta(\mathcal{Y})} \left\langle w, \sum_{s \leq t} k(x^s, x)\mathbf{1}_{y^s} \right\rangle - \frac{1}{\lambda} R(w) \, ,
\end{aligned}
$$

where $\mathbf{1}_y$ is an indicator vector. Thus, the market price update follows a natural kernel-weighted FTRL algorithm, where the learning rate $\lambda$ is the price sensitivity of the market.

## 6   Summary and Future Directions

Motivated by the problem of purchasing data, we gave the first bounded-budget prediction market mechanism that achieves privacy, incentive alignment, and precision (low impact of privacy-preserving noise the predictions). To achieve bounded budget, we first introduced and analyzed a transaction fee, achieving a slowly-growing $O((\log T)^2)$ budget bound, thus eliminating the arbitrage opportunities underlying previous impossibility results. Then, observing that this budget still grows in the number of participants $T$, we further extended these ideas to design an adaptively-growing market, which does achieve bounded budget along with privacy, incentive, and precision guarantees.

We see several exciting directions for future work. An extension of Theorem 4 where $\mathcal{Y}$ need not be finite should be possible via a suitable generalization of Claim 2. Another important direction is to establish privacy for *parameterized* settings as introduced by Waggoner, et al. [12], where instead of kernels, market participants update the (finite-dimensional) parameters directly as in linear regression. Finally, we would like a deeper understanding of the learning–market connection in nonparametric kernel settings, which could lead to practical improvements for design and deployment.

## Footnotes

[1] For convenience we will assume $C$ is twice differentiable, though this is not necessary.

[2]Intuitively, it is enough for the fee to cover arbitrage amounts in expectation, because a trader must pay the fee to trade before the random noise is drawn and any arbitrage opportunity is revealed.

[3]For instance, if the current price of a security is $0.49$ and a trader believes the true price should be $0.50$, she will purchase a share if the fee is $c < 0.01$. (For privacy, we limit each trade to a fixed size, say, one share.)

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
