[Reviews · NeurIPS 2018]

Reviewer 1



This paper studies prediction markets: mechanisms where a series of buyers arrive and make bets on the outcome of a future event, where the prices of bets at a certain point of time reflect the information gathered so far about this outcome. Previous work by Waggoner et al studies differentially-private prediction markets, where no individual buyer's trade is inferrable from the overall sequence of market prices (this is accomplished by adding some amount of noise to each of the revealed market states). These markets have the downside that the expected cost to the mechanism designer of maintaining this differential privacy is unbounded, growing perhaps linearly in T (number of rounds/buyers). This paper extends the previous work by proposing new mechanisms that have bounded total cost to the mechanism designer. Specifically, the authors: 1. The authors show that adding an appropriately chosen transaction fee to the mechanism of Waggoner et al causes the total loss of the mechanism to be bounded by O(log^2 T). (Intuitively, the noise added to maintain differential privacy creates a small arbitrage opportunity for each of the buyers; adding a transaction fee on that scale eliminates this). 2. The authors then show how to "amplify" this O(log^2 T)-loss mechanism to construct a mechanism with O(1)-total loss. This works in a way reminiscent of the doubling trick -- after enough buyers have participated in the current mechanism, the mechanism fee restarts with different parameters (i.e. half the transaction fee). I am not very familiar with the peer prediction literature (including the previous paper by Waggoner et al which this paper builds off of significantly), but this seems like a very strong result, addressing perhaps the largest problem with the previous construction for differentially private prediction markets. In fact, it is not at all clear a priori that a differentially private prediction market with loss bounded by a constant necessarily exists (indeed, Cummings et al show that any of a large class of differentially private prediction markets -- ones that just add some amount of noise to the market states -- must necessarily have unbounded loss). The paper is well-written and easy to read. The proofs in the main body of the paper appear correct (although I have not checked the proofs in the appendix).

Reviewer 2



Main ideas of the submission: This paper studies the problem of running a prediction market to crowdsource data while providing differential privacy guarantees. This problem has previously been studied by Waggonner et al. NIPS 2015 and Cummings et al. EC'16. Waggoner et al. provided a method for purchasing data while guaranteeing differential privacy. Cummings et al. show that one draw back of any differentially private mechanism is that as the number of participants grows the differential privacy gaurantee causes the mechanism to lose on the total profit it can achieve. In this paper the authors propose a way to resolve this issue. In particular, if the mechanism charges each participant a fee for participation, then that can be used to bound the amount of money the mechanism pays. The authors provide a mechanism that pays O((log T)^2) where T is number of participants. They further improve this result using a doubling trick often found in online learning. Using this trick, they can provide a mechanism that spends a constant amount of money while maintaining differential privacy. Quality: The paper is fairly well written and explains the main proof ideas clearly. Originality: While the ideas used in this paper are not new, the authors show conclusively how they work in this setting. Significance: The paper solves an open question posed by a previous paper and provides a strong conclusive answer to the problem.

Reviewer 3



The paper addresses the design of private prediction markets that allow actors to participate in a market to improve accuracy of predictions while preserving privacy. Privacy is introduced by adding noise to the revealed market state, so that past states cannot be used to infer true values. In past work [8] it has been shown that differentially privacy can be guaranteed only in the case of unbounded loss. This paper offers an extension of the model suggested in [8], that the introduction of a transaction fee would alleviate the situation. The present paper presents this model with transaction fees and shows that with transaction fees, privacy can be ensured in the bounded loss case. The authors do not clearly explain the prediction markets model. For a reader unfamiliar with this model it is extremely difficult to follow, especially since many financial terms are unnecessarily used. The authors don’t make it clear how Section 5 fits within the rest. The proof sketch in this section is quite informal and doesn’t give any intuition into the situation.